# Participatory Co-Design and Evaluation of a Novel Approach to Generative AI-Integrated Coursework Assessment in Higher Education

**DOI:** 10.3390/bs15060808

**Published:** 2025-06-12

**Authors:** Alex F. Martin, Svitlana Tubaltseva, Anja Harrison, G. James Rubin

**Affiliations:** 1Department of Psychological Medicine, Institute of Psychiatry, Psychology and Neuroscience, King’s College London, London WC2R 2LS, UK; anja.harrison@kcl.ac.uk (A.H.); gideon.rubin@kcl.ac.uk (G.J.R.); 2School of Liberal Arts, Richmond American University London, London W4 5AN, UK; tubalts@richmond.ac.uk

**Keywords:** generative artificial intelligence, pedagogy, postsecondary, curriculum design, student learning, qualitative

## Abstract

Generative AI tools offer opportunities for enhancing learning and assessment, but raise concerns about equity, academic integrity, and the ability to critically engage with AI-generated content. This study explores these issues within a psychology-oriented postgraduate programme at a UK university. We co-designed and evaluated a novel AI-integrated assessment aimed at improving critical AI literacy among students and teaching staff (pre-registration: osf.io/jqpce). Students were randomly allocated to two groups: the ‘compliant’ group used AI tools to assist with writing a blog and critically reflected on the outputs, while the ‘unrestricted’ group had free rein to use AI to produce the assessment. Teaching staff, blinded to group allocation, marked the blogs using an adapted rubric. Focus groups, interviews, and workshops were conducted to assess the feasibility, acceptability, and perceived integrity of the approach. Findings suggest that, when carefully scaffolded, integrating AI into assessments can promote both technical fluency and ethical reflection. A key contribution of this study is its participatory co-design and evaluation method, which was effective and transferable, and is presented as a practical toolkit for educators. This approach supports growing calls for authentic assessment that mirrors real-world tasks, while highlighting the ongoing need to balance academic integrity with skill development.

## 1. Introduction

Recent advances in generative artificial intelligence (AI), powered by large language models, present opportunities and challenges for assessment in higher education. AI is now widely used across sectors including health, industry, and research ([20]; [25]), and is permanently reshaping the nature of academic tasks. In educational settings, AI has already shown potential to support learning by providing personalised feedback, scaffolding writing processes, and automating routine tasks ([15]; [24]). Interest in the role of AI in education has accelerated rapidly in recent years ([24]), with growing attention being paid to its implications for assessment and feedback practices (e.g., [13]; [29]). In this study, we extend this literature by evaluating a novel assessment design that contrasts different modalities of AI use, providing new insight into how AI can be critically and ethically integrated into higher education assessment. Our participatory methodology is transferable to other educational contexts, and we provide practical resources to support educators in adapting this approach.

Initial studies suggest that, while students may benefit from AI-enhanced feedback, overreliance on these tools may undermine opportunities for deep learning and critical engagement ([15]; [26]; [31]; [32]). The integration of generative AI in education also presents challenges. Equity concerns persist, including unequal access to reliable AI tools and the digital skills needed to use them meaningfully ([28]). Academic integrity is also at risk, as AI can be used to ‘cheat’ in ways that evade detection ([30]). Moreover, the use of AI complicates traditional concepts of authorship and scholarship, raising questions about what constitutes independent academic work ([16]; [18]). There are also concerns that critical thinking, a key goal of higher education, could be weakened if students accept AI outputs without careful evaluation ([2]).

In response, there is growing recognition of the need to build critical AI literacy among students and staff. This means not just knowing how to use AI tools, but understanding how they work, the wider impacts they have, and how to assess AI-generated content carefully and ethically ([1]). Developing critical AI literacy is needed to prepare students to be thoughtful, responsible users of AI, and should be built into teaching and assessment strategies.

The overarching aim of this study is to improve the critical AI literacy of postgraduate students and teaching staff through the co-design and evaluation of an AI-integrated written coursework assessment that contrast different AI modalities. In this assessment, students used generative AI tools to draft a blog critically summarising an empirical research article and produced a reflective, critical commentary on the AI-generated content. Specifically, we asked two research questions:Is the AI-integrated assessment acceptable and feasible for students and teaching staff?Can teaching staff distinguish between assessments completed in accordance with the brief and those generated entirely by AI?

Our findings were used to develop practical guidance and a toolkit for educators, support the implementation and iterative improvement of AI-integrated assessments, and contribute to the wider pedagogical literature on assessment in higher education.

## 2. Materials and Methods

This study uses a participatory evaluation approach. Participatory evaluation involves contributors not just as participants, but as co-designers and co-evaluators ([9]), and has been used previously to explore AI-related resources and curriculum development ([8]; [27]). A strength of this approach is its emphasis on different forms of expertise, including lived experience, disciplinary knowledge, and teaching practice, which contribute to the development of assessments that are both grounded and relevant.

The protocol is available at OSF Registries: doi.org/10.17605/OSF.IO/JQPCE. We used the Guidance for Reporting Involvement of Patients and the Public short form (GRIPP2-SF) checklist to report involvement in the study (reported in Appendix A; ([23]). This study was approved by the Research Ethics Panel of King’s College London (LRS/DP-23/24-42387; 27 June 2024).

This study involves twelve participants from the 2023–24 cohort of a postgraduate course at the Institute of Psychiatry, Psychology, and Neuroscience, King’s College London, a Russell Group university in the United Kingdom. The student cohort comprised approximately 30 individuals. Most were in their early twenties and had entered the MSc programme directly after completing their undergraduate studies, with around one in six being mature students returning to education after spending time in the workforce. A very small number were men. Approximately one-third were UK home students, while two-thirds were international, the majority of whom were from East Asia.

Eight students and four members of the teaching team took part in the study. The teaching staff included a Teaching Fellow, a Lecturer, and two Research Associates. All participants had recently completed or marked a summative assessment within the course. We considered the sample size to be adequate given the small cohort, the participatory nature of the research, and the principle of information power, which suggests that the more relevant information the sample holds, the lower the sample size that is needed ([19]). In this study, participants were well positioned to inform the evaluation, having first-hand experience with the assessment and its development. They brought a range of expertise and experience with AI, from high digital literacy to limited prior use, as well as strengths in academic writing and assessment design. This ensured that the participatory methods supported shared ownership, practical relevance, and opportunities for innovation.

### 2.1. Stage 1: AI-Integrated Assessment

The research team collaborated with other members of the course’s teaching team to adapt an existing summative assessment already embedded in the curriculum. This assessment required students to write a blog post summarising and critically appraising an empirical research article on mental health. Framed as an authentic assessment, the task included the potential for selected blogs to be published on science communication platforms.

We used the Transforming Assessment in Higher Education framework developed by AdvanceHE to guide our approach to integrating generative AI tools into this assessment ([12]). The framework highlights the need for assessments that are authentic, inclusive, and aligned with learning outcomes, emphasising the importance of involving students in the assessment development process. This emphasis aligned with our approach of integrating AI tools to reflect real-world practices and to develop critical AI literacy.

Under the revised assessment approach, students were asked to use two AI tools to assist with drafting a blog based on an empirical article. The written assessment consisted of three components:Two AI-generated blog drafts using two AI tools.A final blog that combined the strongest elements of the AI outputs with the student’s own revisions and original contributions, assessed for the accurate and critical appraisal of the empirical article.A commentary critically reflecting on the AI-generated content and explaining the rationale for revisions made, assessed for the depth of critical and ethical reflection.

The marking matrix was revised to retain the use of a standard critical appraisal checklist for assessing students’ understanding of the empirical article, alongside the programme-wide marking framework (stage 2). New criteria were introduced to evaluate students’ critical engagement with AI-generated content (stage 3). The adapted format built on the existing learning outcome of critically appraising empirical research, extending it to assess students’ ability to reflect on the role of AI in academic work, apply subject knowledge to evaluate AI outputs, and make informed editorial decisions.

### 2.2. Stage 2: Assessment Trial

All participants were invited to take part in a trial of the adapted assessment. They first attended a workshop designed to support students in their AI-assisted assessment. Microsoft Copilot, in both balanced and precise modes, was the mandated generative AI tool, selected for its free availability for the participants (ensuring equitable access) and to allow for direct comparisons between model outputs. While Copilot was used in this instance, the assessment was designed to be transferable to other AI tools.

The workshop was delivered in four stages. The first introduced Copilot’s core functions, including its strengths, limitations, and examples of effective prompt writing. In the second stage, students practised drafting prompts and used the AI models to generate and revise a mock blog post. The final two stages drew on Gibbs’ Reflective Cycle to guide structured learning ([10]). In stage three, students critically appraised an AI-generated blog and compared the outputs produced by the two Copilot modes. This exercise supported a deeper understanding and analysis of AI-generated content. In the final stage, students reflected on their use of AI and developed an action plan for how they would apply AI tools in future academic work. This reflection aimed to consolidate learning and promote ethical, informed use of generative AI tools.

Feedback on the workshop was collected through qualitative discussions at the end of the session and a short survey. The survey included a Likert-scale question assessing whether the workshop would help students complete the assessment (responses: yes, somewhat, no) and two free-text questions: “What did you learn from the workshop?” and “What was missing from the workshop that would help you feel more prepared for the pilot assessment?”

Student participants were then randomly allocated to one of two groups. Those in the ‘compliant’ group were instructed to follow the coursework brief precisely, using the designated AI tools as directed. Students in the ‘unrestricted’ group were given freedom to complete the assessment by any means, including generating the entire submission using AI tools. They were encouraged to be creative and to push the boundaries of the process. Teaching staff participants were asked to mark the submitted assessments and provide written student feedback using the adapted marking matrix. They also indicated whether they believed the student had completed the assessment as instructed (were in the ‘compliant’ group) or had been in the unrestricted group.

### 2.3. Stage 3: Evaluation

Participants participated in an iterative process of reviewing and refining the assessment materials, including the workshop content, coursework brief, and marking matrix.

To explore the feasibility, acceptability, and perceived integrity of the AI-integrated assessment approach, we conducted a series of semi-structured focus groups with students and individual interviews with teaching staff. This format was chosen to accommodate participant preferences and availability, while also helping to reduce power imbalances by providing students with a peer-supported setting in which to reflect on an assessment co-designed with researchers who were also their course instructors.

The discussion guides are reported in Appendix A and at the Open Science Framework project: osf.io/ctewk/. They were developed to address the study’s research aims and to capture experiences across both groups regarding their engagement with generative AI in the context of assessments. Both focus groups and interviews lasted from approximately 45 to 60 min and were structured in two parts: the first explored participants’ existing knowledge of generative AI and their experiences of completing or marking the assessment; the second addressed their reflections on the assessment design and its potential for future implementation. In addition, we explored perceptions of ‘cheating’ in the assessment, including whether students in the compliant and unrestricted groups felt they had met the intended learning outcomes and whether staff felt able to distinguish between the two groups. Particular attention was paid to whether the approach supported intended learning outcomes and provided a fair measure of student performance.

We also asked questions about the initial training workshop as part of the interviews. This feedback was reviewed alongside data from the survey questions completed by participants after the workshop and was used to revise and improve the training content.

Focus groups and interviews were conducted via Microsoft Teams. Thematic analysis was led by one researcher (AFM), following [5]’s ([5]) approach, including familiarisation with the data, initial coding, theme identification, and iterative theme refinement. Analyses was performed separately for students and teaching staff. Emerging themes were reviewed and refined through discussion within the research team and with participants who participated in subsequent workshops.

In addition to qualitative comparisons, we conducted a statistical analysis to compare how successful markers were at identifying assessments written by students in either the compliant or unrestricted groups. Given the small sample size and expected cell counts of below five, we used Fisher’s exact test rather than the chi-square approximation ([14]), calculated using base R ([21]).

We held co-design workshops with students and teaching staff to further refine the assessment brief and marking matrix, respectively. The think aloud technique was used ([6]; [22]), whereby each section of the assessment materials was reviewed in turn. Participants took part in a facilitated group discussion, voicing their thoughts, suggestions, and reactions in real-time as they engaged with the materials. Data saturation was considered to have occurred when no further substantial changes were proposed by the participants. Two workshops were held with students and one with teaching staff, which likely reflects the fact that more extensive feedback had already been gathered from teaching staff during earlier individual interviews and incorporated into the materials prior to the workshops.

Feedback gathered during these sessions was used to inform revisions to the assessment materials. We documented this process using a Table of Changes (ToC) from the Person-Based Approach using the MoSCoW method, a prioritisation framework used to collaboratively decide which features, changes, or recommendations should be implemented [must, should, could, would like] ([4]). We also used a Custom GPT using GPT-4-turbo, which allows for the creation of a personalised version of ChatGPT-4 tailored to specific tasks or knowledge domains, to review the final materials for accessibility and readability.

## 3. Results

### 3.1. Assessment Materials and Learning Outcomes

The modifications made to the assessment materials are summarised in the ToC (Table 1, Table 2, Table 3 and Table 4).

Feedback on the co-designed assessment materials produced using the Custom GPT indicated that, while both the assessment brief and marking matrix were generally well-structured and aligned with learning outcomes, several refinements could improve readability and accessibility. These included ensuring consistency of language and tone, using bullet points and clearer formatting to support navigation, and clarifying instructions around AI tool use and submission structure. Minor revisions were recommended to the learning outcomes and the reflection criteria to enhance alignment with marking expectations.

The final versions of the assessment materials (workshop proformas, assessment brief, and marking matrix) and the amendments recommended by ChatGPT are included in Appendix A and at the Open Science Framework project: osf.io/ctewk/.

Feedback on the learning outcomes was generally positive or neutral, with no negative responses offered. Students and teaching staff appreciated the inclusion of learning outcomes and found them helpful for understanding the purpose of the assessment. Some suggested making the link between the learning outcomes and the specific assessment tasks more explicit to improve alignment and clarify expectations. The major change that emerged from all feedback sources was the need to communicate that critical appraisal of the original empirical article is as important as the appraisal of the ability of AI to generate seemingly useful content. One student noted that engaging with the AI output highlighted inaccuracies, such as fabricated participant details, which prompted them to critically verify the content against the original source. This process, while demanding, was seen as intellectually valuable: “It forces you to actually figure out whether you’re critically appraising the critical appraisal.”(S5) Another contributor reflected on the need to distinguish between assessing AI literacy and assessing critical thinking (S2), suggesting that the learning objectives should clearly indicate which of these skills is being prioritised. This feedback informed revisions to the assessment brief and the revised learning outcomes.

Our revised learning outcomes became the following:
Critical appraisal: Students will demonstrate the ability to critically appraise academic content by the following:Evaluating an empirical research article using an established critical appraisal checklist.Assessing the accuracy, relevance, and limitations of AI-generated content in relation to the original empirical article.Comparing outputs from different AI tools, identifying their strengths and weaknesses in academic content generation.Generative AI literacy: Students will develop foundational AI literacy by using generative tools to support scientific blog writing. They will demonstrate an understanding of AI’s capabilities and limitations, including the ability to identify common errors such as fabrication or hallucination.Editorial and reflective judgement: Students will apply editorial judgement to revise AI-generated content, integrating critical analysis and original insight. They will reflect on their use of AI tools and articulate the rationale for content modifications in alignment with accuracy, academic standards, and ethical considerations.


### 3.2. Feasibility, Acceptability, and Integrity

Table 5 and Table 6 present summaries of the key findings and illustrative quotes from the thematic analysis of the focus groups and interviews.

Student feedback highlighted that, while AI tools could streamline aspects of the writing process, they did not reduce workload due to the effort required to refine outputs. Perceptions of feasibility, acceptability, and integrity varied, with students valuing the opportunity to build critical thinking skills, but also expressing concerns about fairness, skill development, and, notably, ownership of their work. Some viewed equitable access and thoughtful integration of AI to be particularly important for maintaining academic standards. Teaching staff found the assessment structure clear, although marking was initially time-intensive because of the dual task of evaluating both AI and student contributions. Efficiency improved with familiarity, and staff recognised the assessment’s potential to support critical engagement. While challenges remained in distinguishing AI-generated from student-authored content, most staff endorsed transparent and pedagogically grounded use of AI in academic settings.

Students in the unrestricted group found that using AI to complete the entire assessment was challenging, with outputs, particularly the reflective commentary, requiring substantial oversight and correction. Some spent a similar amount of time on the task as those in the compliant group, while others felt they used somewhat less. Most felt they had achieved the intended learning outcomes due to the time spent checking, appraising, and reflecting on the AI-generated content.

Assessment marks ranged from 35 (fail) to 78 (distinction). For most assessments, marks from different markers fell within a ten-point range, but for one assessment, scores ranged more widely (from 58 to 78). Markers correctly identified 6 out of 14 students in the compliant group (42.9%) compared to 3 out of 6 in the unrestricted group (50.0%). Fisher’s exact test produced an odds ratio of 0.75, *p* = 1.00, indicating that marker accuracy did not differ meaningfully between the groups. Markers’ views on identifying students in the unrestricted condition were polarised: some reported having no clear sense, while others felt very confident that they could recognise AI-generated submissions. However, these subjective impressions were not reflected in their actual ability to accurately distinguish between the groups.

## 4. Discussion

The findings from this study add to the nascent body of literature that highlights the dual role of AI-integrated assessments as tools for digital literacy and as mechanisms for reflective, critical pedagogy. The blog format provided a unique opportunity for students to practise public-facing, accessible academic writing, aligning with real-world expectations in science communication. The pilot findings show that students found the approach to be feasible and helpful for developing critical skills, although engaging with AI outputs was perceived to increase workload. Teaching staff initially found marking more demanding and had limited success distinguishing unrestricted AI-generated content, but valued the assessment’s potential to promote ethical and critical AI use.

A key success of the project was the development of students’ critical AI literacy, with findings suggesting that the blog assessment promoted active engagement with AI outputs. Students were required to critique AI-generated content, identify inaccuracies, and justify their editorial decisions. This process appeared to encourage deeper critical engagement and helped students to view AI as a tool requiring human oversight rather than as a source of ready-made answers. However, some students may have used AI to support parts of the evaluative process itself, for example, by prompting AI to critique its own outputs, blurring the boundary between human and AI intervention. This challenge is prompting the development of pedagogical tools to enhance deeper engagement with AI content, including a revised version of Bloom’s Taxonomy ([11]). In our study, students in the unrestricted group reported limited success when attempting to outsource critical reflection and revision entirely to AI, reporting that human oversight remained essential to complete the task successfully. This supports [11]’ ([11]) observation of AI as a cocreator, where students collaboratively refined, challenged, and integrated output. Nonetheless, the timing and degree of human input will vary between students, highlighting the need for structured scaffolding to support meaningful engagement with AI whilst safeguarding academic skill development.

The requirement to compare outputs across different AI models also supported the development of critical evaluation skills, as students reflected on the variability and limitations of AI-generated content. Importantly, these findings address concerns raised during the qualitative evaluation and reflected issues highlighted in previous research, such as that overreliance on AI could undermine opportunities for deep learning and reflective practice ([15]; [17]; [31]). These findings align with recommendations that AI in education must go beyond functional skills to include AI literacy, as well as active learning skills and metacognition ([1]).

Beyond promoting critical engagement with AI outputs, this study also highlights strategies for maintaining assessment integrity and supporting academic skill development. Teaching staff expressed concerns that AI use could make it harder to distinguish original work from AI-generated content. This echoes broader challenges in the literature, where AI use may complicate traditional definitions of scholarship and independent academic work ([16]; [18]; [30]). Although markers were sometimes confident, their accuracy in identifying AI-reliant submissions from those that were compliant with the assessment instructions was poor. This is likely because students in the unrestricted group generally described a similar editorial process to those in the compliant group. Nevertheless, the assessment’s structure, requiring critical appraisal of the empirical article, critique of AI outputs, evidence of revision, and transparency may have helped to mitigate these risks, although this needs further testing.

By embedding critical evaluation and editorial judgement, the assessment addressed concerns that AI could weaken core academic skills such as critical thinking and reflective analysis ([2]). One key challenge identified by participants was that the focus on evaluating AI-generated content risked overshadowing the critical appraisal of the empirical article itself. In response, the final co-produced brief more clearly separated and emphasised both components and better balanced the dual aims of the task. Maintaining this balance will be essential in future implementations to ensure that the assessment remains both authentic and educationally robust. Students also recognised that genuine engagement, not uncritical acceptance of AI outputs, was needed to meet the learning outcomes. However, the extent to which students internalised critical evaluation versus simply complying with task requirements remains unclear. Future studies could explore students’ metacognitive strategies and critical reasoning during AI use through longitudinal or think-aloud methodologies ([6]; [22]). Overall, the findings suggest that carefully designed AI-integrated assessments can uphold academic integrity while supporting the development of essential academic competencies.

Involving students and teaching staff in the co-design and evaluation process was central to developing an assessment that was authentic, feasible, and acceptable. The participatory approach drew on academic, pedagogical, and lived experience to shape the teaching workshop and assessment materials, helping us spot practical challenges early and promote shared ownership of the development of the assessment ([9]; [27]). This aligns with broader calls for more inclusive, responsive, and transparent innovation in educational assessment ([3]; [12]). However, participatory approaches also carry limitations, including potential power imbalances between participants and researchers, risks of tokenism, and the possibility of over-relying on stakeholder input to the detriment of expert judgement. Future research should continue to embed participatory evaluation while remaining mindful of these challenges to ensure AI-integrated assessment remains student-centred and pedagogically sound.

Several limitations of this study should be acknowledged. First, students were not involved in the initial design phase of the assessment, falling short of authentic co-production ([7]). Although this was partly mitigated through later participatory evaluation, involving students earlier could have strengthened the creativity, relevance, and ethical responsiveness of the assessment. Second, qualitative feedback was collected following the initial pilot rather than after a full module-wide rollout. As such, findings may reflect early impressions rather than longer-term engagement. However, this timing allowed for immediate adjustments and iterative revisions of the assessment materials. Third, this study was conducted within a single institutional setting with a small cohort and an ensuing small sample size, which limits the generalisability of the evaluation findings to other universities or international contexts with different AI access, policies, and pedagogical cultures. However, this study did not aim for statistical generalisability, but rather aimed to explore the feasibility and acceptability in context, using participatory methods grounded in information power ([19]). Our broader goal was to model a co-design and evaluation approach that is transferable and could be adapted to different educational settings. The resulting assessment toolkit supports wider applications, helping educators adapt AI-integrated assessments to their own institutional and disciplinary contexts.

## 5. Conclusions

This study explores the participatory development and evaluation of a generative AI-integrated assessment in postgraduate education. The participatory methods used were effective in shaping an assessment that was both feasible and meaningful. A practical toolkit was produced to enable educators to apply similar co-design and evaluation processes within their own teaching contexts. Findings from this pilot evaluation suggest that integrating AI into assessments can promote both technical fluency and ethical reflection when scaffolded appropriately. Students engaged critically with AI outputs, while teaching staff recognised the potential for supporting critical thinking and maintaining academic integrity. Our approach supports growing calls for authentic assessments that mirrors real-world tasks, particularly in professions where AI is becoming more common. However, there remains a tension between preserving academic integrity and using AI to support skill development. Future iterations must continue to navigate this balance carefully, ensuring that critical engagement and ethical practice are at the core of AI-integrated learning.

## Figures and Tables

**Table 1 behavsci-15-00808-t001:** Table of changes: Summary of ‘must’ changes to assessment materials with feedback sources.

Component	Source	Feedback	Proposed Change	Rationale	Agreed Change
Assessment Brief	Staff workshop	Instructions lacked clarity in appraising both the original article and the AI’s interpretation	Have a separate section that states expectations for appraising the empirical article	Ensures students understand and address both components of the assessment	Clarification added under Instructions section Critical appraisal learning outcome split into three exclusive sections Additional content added at top of Tips section describing the need to critically understand the empirical article
Assessment Brief	Student workshop	Instructions for the reflection part of the assessment was somewhat unclear	Add more detailed guidance and reflective models/frameworks	Helps students structure their critique and understand expectations	4. Reflective models are recommended under Instructions section
Assessment Brief	Student workshop	Referencing format unclear	Specify referencing style in blog guidance	Addresses AI limitations in citation generation	5. Clarity of referencing style added to Blog Guidance section
Assessment Brief	Student workshop	Students appreciated learning outcomes, but overlooked critical appraisal	Emphasise critical appraisal more clearly throughout and link to learning outcomes	Reinforces key educational focus and clarifies expectations	6. Critical appraisal learning outcome split into three exclusive sections

**Table 2 behavsci-15-00808-t002:** Table of changes: Summary of ‘should’ changes to assessment materials with feedback sources.

Component	Source	Feedback	Proposed Change	Rationale	Agreed Change
Assessment Brief	Student workshop	Word count guidance inconsistent	State word count for each section and total limit clearly	Reduces confusion and supports appropriate planning	7. Wordcounts are provided for the total and for each part of the report under the Instructions section
Assessment Brief	Student workshop	Value of tips section highlighted	Retain and expand guidance on recognising AI hallucinations and faults	Reinforces critical AI literacy and practical assessment skills	8. Added to the learning outcomes 9. Added to ‘risks’ in the Background section 10. Expanded the Tips section
Assessment Brief	Staff workshop	Students may not fully demonstrate critical engagement with AI-generated content	Require students to highlight changes made to AI-generated content and include in an appendix	Encourages transparency and supports assessment of student input	11. Added to the Instructions for coursework section
Marking Matrix	Staff workshop	Lack of marker guidance for grade boundaries	Provide examples of high- and low-quality work	Improves marker confidence and consistency	12. Included examples from high and grade bounds
Marking Matrix	Staff interviews	Time consuming to check against academic articles	Provide checklist for articles of key methods, results, and conclusions	Improves marker confidence and consistency	13. Included checklist for empirical articles
Training Workshop	Student workshop	Ethical use of AI is not well understood	Include declaration forms, videos, and exemplar prompts in the training materials	Supports transparency and encourages ethical practice	14. Added an interactive brainstorming activity on the ethical use of AI, tailored to the specific context of the assessment 15. Incorporated group breakout discussions to support peer learning and reflective engagement
Training Workshop	Student workshop	Students requested more support with AI prompting	Include dedicated time in the workshop focused on writing effective AI prompts	Builds critical AI literacy and confidence in tool use	16. Integrated into revised two-part workshop design, with optional homework to support independent learning

**Table 3 behavsci-15-00808-t003:** Table of changes: Summary of ‘could’ changes to assessment materials with feedback sources.

Component	Source	Feedback	Proposed Change	Rationale	Agreed Change
Assessment Brief	Student workshop	Background section was appreciated, but some found it overwhelming	Use bullet points instead of paragraphs	Improves accessibility and reduces cognitive load	17. The background and relevance section has been streamlined and uses bullet points to break down heavier text
Assessment Brief	Student workshop	Understanding the purpose improved engagement	Highlight the brief rationale or real-world relevance in the assessment introduction	Enhances motivation and situates learning in context	18. Extended the ‘why blogs’ and ‘why generative AI’ sections of the Background section
Assessment Brief	Staff interviews	Complexity of articles may limit scope for critical appraisal	Use more complex articles with intentional limitations or errors that are not highlighted by the study authors	Allows for students to demonstrate deeper critical thinking and analytical skills	19. Added to the Instructions for coursework section
Training Workshop	Student workshop	Students need more practical support in using AI	Offer two workshops at different levels across the programme	Caters to varying levels of familiarity and ensures accessible skill development	20. Revised workshop structure to offer two scaffolded sessions: - Foundations 1: Covers ethics and core AI concepts, with optional homework to reinforce learning - Foundations 2: Focuses on practical assessment-based activities, including a review of the King’s AI coversheet
Training Workshop	Training session survey	Limited prior understanding of how to use AI; prompting guidance was especially valuable	Expand workshop content on prompt engineering, including examples and guided practice sessions	Builds foundational skills for effective AI use and supports equitable engagement with the assessment format	21. Included in the scaffolded workshop structure and aligned with core learning outcomes and feedback priorities

**Table 4 behavsci-15-00808-t004:** Table of changes: Summary of ‘would like’ changes to assessment materials with feedback sources.

Component	Source	Feedback	Proposed Change	Rationale	Agreed Change
Marking Matrix	Staff workshop	Difficulty in marking AI vs. student input	Integrate AI evaluation criteria into the main marking matrix	Reduces marker cognitive load and reflects integrated learning outcomes	22. Out of the scope of the pilot as it addresses the programme-wide marking matrix
Training Workshop	Staff workshop	Markers need guidance on evaluating AI-assisted work	Provide training sessions specifically for staff assessing AI-integrated submissions	Supports consistency and confidence in marking across staff teams and supports new format adoption	23. Marked as a priority for inclusion before the next phase of the trial’s implementation

**Table 5 behavsci-15-00808-t005:** Key findings and illustrative quotes from students.

Theme	Key Finding	Contributor Quotes
Students		
Feasibility: efficiency vs. effort	Many students found that generative AI tools streamlined the writing process, but did not necessarily change the overall workload, particularly when refining AI outputs.	“[the assessment] took about the same amount of time that it would normally … that that was a big shock to me!”(S4) “It forces you to figure out whether you’re critically appraising the critical appraisal, but you think it’s given. So it is a bit, it is more work in that sense.”(S3)
Feasibility: user variability	Students highlighted varying levels of success when using AI, depending on their AI literacy and academic skills.	“I couldn’t get it to be any longer, no matter how many times I prompt it … so that’s probably user error.”(S4) “I started off using AI, but then I found that it was giving very predictable answers … so I then added a lot to that and brought in my own experience into it.”(S6)
Acceptability: learning enhancement	Some students saw the integration of AI as a valuable tool for building critical thinking and learning how to appraise content more deeply.	“It would improve students’ critical thinking abilities if they don’t have to focus as much on things like grammar … they can focus more on the topics at hand, gaps in research.”(S4) “It can help them do the basic task, learn from it, but at the same time still require them to give some critical viewpoints … I only see positivity.”(S1)
Acceptability: skill development	Others raised concerns that reliance on AI might compromise the development of core academic skills.	“It would be sad to lose [the skill of writing] an essay with good grammar … losing some of the core skills of sort of being a student at university”(S3) “Are we learning anything, or are we just asking [AI]?”
Integrity and fairness: ethics and ownership	Some students expressed discomfort about using AI-generated content, especially when it felt detached from their intellectual effort.	“It kind of alienated me from the work … I had to keep going back to be like, did I say this?”(S4) “I wasn’t exactly tired after I’d finished it because it did a lot of the work, but like, it was like, I don’t read it and feel like any kind of pride or any kind of like ownership of it in any way.”(S6)
Integrity and fairness: equity and academic standards	Some responses reflected concerns around integrity and fairness of using AI in assessment generally.	“It’s going to start feeling like we’re being assessed on how well we can use AI.”(S5) “It doesn’t make me any more tempted to use it in university assignments, because I feel like I’d come out of it with a degree that I didn’t really feel like was mine.”(S6)
Integrity and fairness: equity and academic standards	Some were more optimistic about AI in the mock assessment.	‘The problem happens when some people in the classroom are relying on AI, some people are not. So then it creates a kind of disparity. But if everyone is using it then … I would feel that would be the most fair updater of ability because then everyone has the same resource.”(S1)

**Table 6 behavsci-15-00808-t006:** Key findings and illustrative quotes from teaching staff.

Teaching Staff		
Feasibility: complexity of marking	Most staff reported that while the structure was clear, marking took longer due to the added components and the unfamiliar task of evaluating AI use.	“The first one took me two hours … I had to read the paper so that I wasn’t taking into account how long it takes.”(T2) “You have a blog, and a commentary, it doubles the work.”(T1)
Feasibility: familiarity improved efficiency	Once familiar with the marking expectations, staff found the process more manageable.	“Afterwards, it took about an hour…. once I figured out what I was doing.”(T2) “I take more time in the first in the first or second, but then generally the blog part it was quite easy to mark.”(T3)
Acceptability: critical thinking	Staff saw potential in the assessment for encouraging deeper student engagement and critical evaluation.	“Incorporating AI into assessments is interesting, because … you still need to think and you still need to produce something.”(T1) “I think that’s good for some students. The work I read [was] really interesting. Some students did read well and add their critics. I’m really impressed about … how they learn from that.”(T3)
Acceptability: undermining learning	There were worries that integration of AI might limit the development of core academic skills.	“You ask ChatGPT, it removes the painful experience [of writing well], but I think … we need that struggle.”(T4) “If we just rely on chat GPT writing things for us and then thinking for us, we will lose the ability to be creative ourselves, because I feel like you learn by doing.”(T2)
Integrity and fairness: evaluation	Staff expressed uncertainty about distinguishing between student-authored and AI-generated content in more formulaic sections, but many were confident that they could identify inappropriate use of AI in the assessment.	“AI has done a pretty good job describing the methodology.”(T1) “You can tell when you read, some sections of the blog that can also be impersonal, like the methods and results … The personal perspective that’s easier to judge because you can kind of sense the touch human touch.”(T4) “There’s no critical thinking because obviously the blog has been written by AI, so there’s automatically no critical thinking there, and they didn’t particularly try to add more on their own behalf.”(T1)
Integrity and fairness: integration	Despite concerns, staff generally recognised the inevitability of AI in education and advocated for proactive adaptation.	“It helps people understand and think critically about [AI] use.”(T2) “We might as well embrace it and … teach people how to [use it properly].”(T4)

## Data Availability

Given the participatory nature of this research and the potential for participants to be identifiable in full transcripts, the data cannot be shared with third parties.

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
