# Peer review of "Participatory Co-Design and Evaluation of a Novel Approach to Generative AI-Integrated Coursework Assessment in Higher Education"

_behavsci, 2025, doi:10.3390/bs15060808_

Round 1
Reviewer 1 Report
Comments and Suggestions for Authors
Thank you for the opportunity to review this interesting paper. My impression is that much of the discussion around AI in assessment thus far has focused on the potentials (and potential merits/demerits) of deploying AI in traditional, non-AI-integrated assessment tasks. The distinctiveness of this study lies in its evaluation of different modalities of AI use within an assessment context that is already AI-integrated. The paper’s demonstration of and advocacy for the participatory evaluation approach is also significant given the current knowledge deficit regarding students’ AI-related beliefs and practices.
I found the paper highly readable, well organised, and sound in its observations. I have just a few small suggestions which I would like to see addressed if there is space to do so.
- Participatory evaluation is an excellent method to adopt here, but it would also be worth acknowledging its limitations/drawbacks, e.g. effect of power imbalances among participants and between participants and researchers (especially in the case of the individual interviews), potential to be tokenistic (participatory findings not applied substantively) or over-exposed (findings applied excessively, to the point of devaluing expert perspectives).
- I wonder if the labels ‘honest’ and ‘cheating’ are appropriate? They set up a black-and-white distinction which doesn’t seem applicable in an assessment task where AI use is already integrated and mandated. The difference between the two groups is one of DEGREE of AI use (compliant with the instructions vs unrestricted) rather than honesty.
- One key observation from participants is ‘the need to communicate that critical appraisal of the original empirical article is as important as appraisal of the ability of AI to generate seemingly useful content’ (l.226-7). This draws attention to a key challenge in designing tasks like this, which is that the focus on techniques for evaluating the AI itself (AI literacy) can often overshadow the original purpose of applying subject/discipline skills in the critical appraisal of research (academic literacy). A little more discussion/reflection on this point could be useful, including what kind of balance the authors hope to strike in future iterations of this activity.
- I would advise a little more caution in the portrayal of supposedly non-AI components of the activity. For example: ‘Students were required not only to use AI tools but to critique their outputs, identify inaccuracies, and justify their editorial decisions. This process encouraged deeper critical engagement and helped students to view AI as a tool requiring human oversight rather than as a source of ready-made answers’ (l.302-304) – potentially this is true, but it is also quite conceivable - and indeed common, in my experience - that students will outsource these tasks of justifying and critiquing to AI as well (getting AI to evaluate AI). Given this possibility, it might be best not to describe components such as the blog activity unproblematically as mechanisms for human oversight and critical engagement. This is not to deny their value in terms of building AI literacy: it’s just that the timing and degree of human intervention may vary from student to student.
Author Response
Thank you for considering our manuscript for publication in Behavioural Sciences. We appreciate the encouraging and supportive feedback and are grateful to the reviewers for insightful comments that have considerably improved the quality of the manuscript. Here is the response to reviewer 1 (the attached word doc includes clearer formatting):
- Participatory evaluation is an excellent method to adopt here, but it would also be worth acknowledging its limitations/drawbacks, e.g. effect of power imbalances among participants and between participants and researchers (especially in the case of the individual interviews), potential to be tokenistic (participatory findings not applied substantively) or over-exposed (findings applied excessively, to the point of devaluing expert perspectives).
Thank you, we have updated the discussion accordingly (lines 640-645)
‘This aligns with broader calls for more inclusive, responsive, and transparent innovation in educational assessment (Bovill et al., 2016; Healey & Healey, 2019). However, participatory approaches also carry limitations, including potential power imbalances between participants and researchers, risks of tokenism, and the possibility of over-relying on stakeholder input to the detriment of expert judgment. Future research should continue to embed participatory evaluation while remaining mindful of these challenges to ensure AI-integrated assessment remains student-centred and pedagogically sound.’
- I wonder if the labels ‘honest’ and ‘cheating’ are appropriate? They set up a black-and-white distinction which doesn’t seem applicable in an assessment task where AI use is already integrated and mandated. The difference between the two groups is one of DEGREE of AI use (compliant with the instructions vs unrestricted) rather than honesty.
This is an excellent suggestion, we agree that the descriptors ‘compliant’ and ‘unrestricted’ are more accurate and nuanced. We have updated to these descriptors throughout the manuscript.
- One key observation from participants is ‘the need to communicate that critical appraisal of the original empirical article is as important as appraisal of the ability of AI to generate seemingly useful content’ (l.226-7). This draws attention to a key challenge in designing tasks like this, which is that the focus on techniques for evaluating the AI itself (AI literacy) can often overshadow the original purpose of applying subject/discipline skills in the critical appraisal of research (academic literacy). A little more discussion/reflection on this point could be useful, including what kind of balance the authors hope to strike in future iterations of this activity.
This is also an insightful suggestion; this tension was strongly articulated in the participant feedback and we appreciate the opportunity to draw this out more thoroughly in the manuscript. We have added the following to the paragraph where the dual nature of the task is discussed (lines 620-625):
‘By embedding critical evaluation and editorial judgement, the assessment addressed concerns that AI could weaken core academic skills such as critical thinking and reflective analysis (Bittle & El-Gayar, 2025). One key challenge identified by participants was that the focus on evaluating AI-generated content risked overshadowing the critical appraisal of the empirical article itself. In response, the final co-produced brief more clearly separated and emphasised both components and better balanced the dual aims of the task. Maintaining this balance will be essential in future implementations to ensure the assessment remains both authentic and educationally robust. Students also recognised that genuine engagement, not uncritical acceptance of AI outputs…’
- I would advise a little more caution in the portrayal of supposedly non-AI components of the activity. For example: ‘Students were required not only to use AI tools but to critique their outputs, identify inaccuracies, and justify their editorial decisions. This process encouraged deeper critical engagement and helped students to view AI as a tool requiring human oversight rather than as a source of ready-made answers’ (l.302-304) – potentially this is true, but it is also quite conceivable - and indeed common, in my experience - that students will outsource these tasks of justifying and critiquing to AI as well (getting AI to evaluate AI). Given this possibility, it might be best not to describe components such as the blog activity unproblematically as mechanisms for human oversight and critical engagement. This is not to deny their value in terms of building AI literacy: it’s just that the timing and degree of human intervention may vary from student to student.
This is a great point and we have integrated a discussion on (lines 571-585):
‘A key success of the project was the development of students’ critical AI literacy, with findings suggesting that the blog assessment promoted active engagement with AI outputs. Students were required to critique AI-generated content, identify inaccuracies, and justify their editorial decisions. This process appeared to encourage deeper critical engagement and helped students to view AI as a tool requiring human oversight rather than as a source of ready-made answers. However, some students may have used AI to support parts of the evaluative process itself, for example, by prompting AI to critique its own outputs, blurring the boundary between human and AI intervention. This challenge is prompting the development of pedagogical tools to enhance deeper engagement with AI content, including a revised version of Bloom’s Taxonomy (Gonsalves, 2024). In our study, students in the unrestricted group reported limited success when attempting to outsource critical reflection and revision entirely to AI, reporting that human oversight remained essential to complete the task successfully. This supports Gonsalves’ (2024) observation of AI as a cocreator, where students collaboratively refined, challenged, and integrated output. Nonetheless, the timing and degree of human input will vary between students, highlighting the need for structured scaffolding to support meaningful engagement with AI whilst safeguarding academic skill development.’

Reviewer 2 Report
Comments and Suggestions for Authors
The study reports on an innovative approach to evaluating generative AI-integrated educational applications. However, the number of source references in the survey is relatively modest (20 items), the bibliography is relevant, and the references provide a good argumentative background for the research and the publication. The topic specification, design, and evaluation are based on the participatory method, and the indication of acceptance by students and teachers, and the various applications of AI tools is convincing. The correct preparation and exemplary documentation of the study's professional methodological and ethical aspects should be highlighted. Based on the number of participants, the research presents a pilot study. Even if all this cannot be considered representative by definition, it is remarkable in its methodological elements and provides a sufficient basis for scientific analysis. The Materials and Methods chapter offers a well-structured, detailed description of the research process. From a formal point of view, however, the decimal list inserted on the second and third pages is confusing, while the content aspects and highlights are important. The research results were presented in two large (3-5 pages!) tables. In the reader, especially on an online interface, the software raises doubts about how suitable multi-page tables are for comparative analysis of the results on a single screen page. In this regard, striving for a more rational graphic solution would have been advisable, even to allow the authors to fold the tables.
Author Response
Thank you for considering our manuscript for publication in Behavioural Sciences. We appreciate the encouraging and supportive feedback and are grateful to the reviewers for insightful comments that have considerably improved the quality of the manuscript. Here is the response to reviewer 2.
- From a formal point of view, however, the decimal list inserted on the second and third pages is confusing, while the content aspects and highlights are important.
Thank you for pointing this out. We think that this was an error in the formatting of the bulleted lists, which have now been indented on lines 84-87 and 162-168. We will work with the editing team to ensure the formatting is presented clearly in the final proof.
- The research results were presented in two large (3-5 pages!) tables. In the reader, especially on an online interface, the software raises doubts about how suitable multi-page tables are for comparative analysis of the results on a single screen page. In this regard, striving for a more rational graphic solution would have been advisable, even to allow the authors to fold the tables.
Thank you for pointing this out. This highlights an unfortunate limitation of the software, as tables of changes and thematic overview tables are a standard way of presenting these data. To mitigate this, we have split the tables and, if accepted, will work with the production team to ensure each fits to one page. The Table of Changes has been split into four tables (Tables 1-4) based on the MoSCoW priority rating (must, should, could, would like). The thematic overview has been split into student and teacher tables (Tables 5 and 6), and we have removed some of the details from the illustrative quotes so that they each fit on one page.
Reviewer 3 Report
Comments and Suggestions for Authors
This is a highly interesting, timely, and well-written manuscript. The study addresses a critical and evolving challenge in higher education: how to meaningfully integrate GenAI into assessment practices while fostering critical AI literacy among students and staff. By co-designing and evaluating an AI-integrated written coursework task, the authors make a valuable contribution to the emerging discourse on responsible AI use in education. Please see my comments below:
Introduction section - the opening paragraph effectively outlines the relevance of GenAI in the field of higher education assessment. However, I recommend updating and strengthening the literature base to reflect the rapid pace of developments in this field. For instance, the sentence stating that “AI has already shown potential to support learning by providing personalised feedback, scaffolding writing processes, and automating routine tasks” should be supported by more recent and targeted references, ideally from 2025, given the abundance of new studies published on this topic in the past year. Furthermore, the claim that “evidence on how best to use it in teaching, especially assessment design, remains limited” also requires citations. While this may have been the case in earlier stages of AI adoption, 2025 has seen a notable growth in empirical work on the integration of GenAI tools in assessment contexts in higher education. You may want to read and cite recent papers from the journal Assessment & Evaluation in Higher Education, which has published in the past year several papers very relevant to your claims. See for example:
- Usher, M. (2025). Generative AI vs. instructor vs. peer assessments: A comparison of grading and feedback in higher education. Assessment and Evaluation in Higher Education, 1–16. https://doi.org/10.1080/02602938.2025.2487495
- Henderson, M., Bearman, M., Chung, J., Fawns, T., Buckingham Shum, S., Matthews, K. E., & de Mello Heredia, J. (2025). Comparing Generative AI and teacher feedback: student perceptions of usefulness and trustworthiness. Assessment & Evaluation in Higher Education, 1–16. https://doi.org/10.1080/02602938.2025.2502582
* The manuscript lacks clearly stated research questions. While the study’s objectives are generally outlined, articulating explicit research questions would significantly strengthen the structure and focus of the manuscript.
Materials and methods section - I have several concerns regarding the sample and the way data were collected from participants. First, the overall sample size is extremely small - only 12 participants, including just 8 students and 4 teaching staff from a single academic cohort. This limits the generalizability of the findings and constrains the range of perspectives represented.
Second, the manuscript lacks important contextual information about the participants. For a study focused on critical AI literacy and the integration of GenAI tools in assessment, it is essential to report details such as participants’ age, gender, academic background, and especially their prior experience and familiarity with GenAI tools for learning purposes. Such information is necessary to meaningfully interpret how students engaged with the task and reflected on the AI-generated content. Without this context, it is difficult to assess the validity and depth of their responses or to determine the broader applicability of the findings.
Third, the choice to use focus groups with students rather than individual interviews deserves an explanation. Since the AI-integrated task was completed individually, and only eight students were involved, it would have been feasible - and probably methodologically stronger, to conduct individual interviews. Focus groups can sometimes suppress divergent or critical views due to social dynamics. Individual interviews would have allowed for richer insights into each student’s unique experience with the task and the AI tools.
Lastly, although the authors includes a limitations paragraph that appropriately acknowledges the study’s confinement to a single institutional context, they do not address the significant limitation posed by the very small sample size. With only 8 student participants, the findings should be interpreted with considerable caution. I recommend that the authors revise their limitations section to explicitly acknowledge the small sample size and its implications for the validity and broader applicability of their findings.
Results section - In this section, participant quotes are presented – I recommend including basic identifiers alongside each quote. For example, participant codes such as (S1, Male) or (T3, Teaching Staff) to clarify who is speaking. This practice is common in qualitative research. Given the small number of participants, using consistent coding (e.g., S1–S8 for students and T1–T4 for staff) would be sufficient to maintain anonymity while allowing readers to better contextualize the quotes.
Moreover, the paragraph in subsection 3.2 (rows 276-284) regarding marker accuracy and statistical testing requires clarification. It is currently unclear how the statistical comparison was justified and conducted given the extremely small number of cases. With such small cell sizes, the assumptions of the chi-square test are likely violated. Additionally, this statistical test is not mentioned in the Methods section.
Discussion section - this section tends to reiterate results rather than critically interpret them in relation to the broader body of existing literature. There is limited engagement with recent empirical work in the field, particularly studies from 2024–2025 that have explored the integration of GenAI in higher education assessment. As I noted in my comments on the Introduction, this is a rapidly developing area, and the discussion would benefit from a stronger effort to position the study’s contributions within the evolving academic discourse. I encourage the authors to incorporate and reflect on recent and empirical studies that address similar pedagogical, ethical, and assessment-related challenges and innovations. Additionally, a more critical reflection on the implications for practice and policy would strengthen the contribution of the manuscript.
Good luck in revising the manuscript!
Author Response
Thank you for considering our manuscript for publication in Behavioural Sciences. We appreciate the encouraging and supportive feedback and are grateful to the reviewers for insightful comments that have considerably improved the quality of the manuscript. Here is the response to reviewer 3. (the attached word doc includes clearer formatting).
Introduction section - the opening paragraph effectively outlines the relevance of GenAI in the field of higher education assessment.
- However, I recommend updating and strengthening the literature base to reflect the rapid pace of developments in this field. For instance, the sentence stating that “AI has already shown potential to support learning by providing personalised feedback, scaffolding writing processes, and automating routine tasks” should be supported by more recent and targeted references, ideally from 2025, given the abundance of new studies published on this topic in the past year. Furthermore, the claim that “evidence on how best to use it in teaching, especially assessment design, remains limited” also requires citations. While this may have been the case in earlier stages of AI adoption, 2025 has seen a notable growth in empirical work on the integration of GenAI tools in assessment contexts in higher education. You may want to read and cite recent papers from the journal Assessment & Evaluation in Higher Education, which has published in the past year several papers very relevant to your claims. See for example:
- Usher, M. (2025). Generative AI vs. instructor vs. peer assessments: A comparison of grading and feedback in higher education. Assessment and Evaluation in Higher Education, 1–16. https://doi.org/10.1080/02602938.2025.2487495
- Henderson, M., Bearman, M., Chung, J., Fawns, T., Buckingham Shum, S., Matthews, K. E., & de Mello Heredia, J. (2025). Comparing Generative AI and teacher feedback: student perceptions of usefulness and trustworthiness. Assessment & Evaluation in Higher Education, 1–16. https://doi.org/10.1080/02602938.2025.2502582
Thank you for this suggestion. We have updated the introduction to include more recent and targeted references. In addition to those suggested, we have also included:
Education transformation: Strielkowski, W., Grebennikova, V., Lisovskiy, A., Rakhimova, G., Vasileva, T., 2025. AI‐driven adaptive learning for sustainable educational transformation. Sustainable Development. 33, 1921-1947.
Academic authorship and scholarship: Kulkarni, M., Mantere, S., Vaara, E., van den Broek, E., Pachidi, S., Glaser, V.L., Gehman, J., Petriglieri, G., Lindebaum, D., Cameron, L.D., 2024. The future of research in an artificial intelligence-driven world. Journal of Management Inquiry. 33, 207-229.; Luo, J., 2024. A critical review of GenAI policies in higher education assessment: a call to reconsider the “originality” of students’ work. Assessment & Evaluation in Higher Education. 49, 651-664. https://doi.org/10.1080/02602938.2024.2309963.
Critical thinking: Zhai, C., Wibowo, S., Li, L.D., 2024. The effects of over-reliance on AI dialogue systems on students' cognitive abilities: a systematic review. Smart Learning Environments. 11. https://doi.org/10.1186/s40561-024-00316-7; Suriano, R., Plebe, A., Acciai, A., Fabio, R.A., 2025. Student interaction with ChatGPT can promote complex critical thinking skills. Learning and Instruction. 95, 102011.
We have also updated the introduction to provide clarity on the aims of the research (lines 37-60):
‘Recent advances in generative artificial intelligence (AI), powered by large language models, present opportunities and challenges for assessment in higher education. AI is now widely used across sectors including health, industry, and research (McKinsey, 2024; Sun et al., 2023), and is permanently reshaping the nature of academic tasks. In educational settings, AI has already shown potential to support learning by providing personalised feedback, scaffolding writing processes, and automating routine tasks (Kasneci et al., 2023; Strielkowski et al., 2025). Interest in the role of AI in education has accelerated rapidly in recent years (Strielkowski et al., 2025), with growing attention to its implications for assessment and feedback practices (e.g. (Henderson et al., 2025; Usher, 2025). In this study, we extend this literature by evaluating a novel assessment design that contrasts different modalities of AI use, providing new insight into how AI can be critically and ethically integrated into higher education assessment. Our participatory methodology is transferable to other educational contexts and we provide practical resources to support educators in adapting this approach.’
- The manuscript lacks clearly stated research questions. While the study’s objectives are generally outlined, articulating explicit research questions would significantly strengthen the structure and focus of the manuscript.
We have updated the introduction to clarify the study’s aims and questions (lines 78-90)
‘The overarching aim of this study was to improve the critical AI literacy of postgraduate students and teaching staff through the co-design and evaluation of an AI-integrated written coursework assessment that contrast different AI modalities. In this assessment, students used generative AI tools to draft a blog critically summarising an empirical research article and produced a reflective, critical commentary on the AI-generated content. Specifically, we asked two research questions:
- Is the AI-integrated assessment acceptable and feasible for students and teaching staff?
- Can teaching staff distinguish between assessments completed in accordance with the brief and those generated entirely by AI?
Findings were used to develop practical guidance and a toolkit for educators, to support the implementation and iterative improvement of AI-integrated assessment and contribute to the wider pedagogical literature on assessment in higher education.’
Materials and methods section
- I have several concerns regarding the sample and the way data were collected from participants. First, the overall sample size is extremely small - only 12 participants, including just 8 students and 4 teaching staff from a single academic cohort. This limits the generalizability of the findings and constrains the range of perspectives represented
Thank you for this reflection. We were not aiming for generalisability in the evaluation of this qualitative, co-creation-based pilot study and we anticipated a small sample size. However, we have now included text regarding the sample size to provide some reassurance on this point in the Materials and Methods (lines 135-143):
‘We considered the sample size adequate given the small cohort, the participatory nature of the research, and the principle of information power, which suggests that the more relevant information the sample holds, the lower the sample size needed (Malterud et al., 2016). In this study, our participants were well positioned to inform the evaluation, having first-hand experience with the assessment and its development.’
We have addressed this important point further in response to #12 below.
We did however aim for generalisability of the method itself to different academic contexts, and we have attempted to make this clearer in the revised abstract (lines 22-24):
‘Findings suggest that, when carefully scaffolded, integrating AI into assessment can promote both technical fluency and ethical reflection. A key contribution of this study is its participatory co-design and evaluation method, which was effective and transferable, and is presented as a practical toolkit for educators.’
Aims (lines 88-90):
‘Findings were used to develop practical guidance and a toolkit for educators, to support the implementation and iterative improvement of AI-integrated assessment and contribute to the wider pedagogical literature on assessment in higher education.’
And the conclusion (lines 683-686):
‘This study explored the participatory development and evaluation of a generative AI-integrated assessment in postgraduate education. The participatory methods used were effective in shaping an assessment that was both feasible and meaningful. A practical toolkit has been produced to enable educators to apply similar co-design and evaluation processes within their own teaching contexts. Findings from this pilot evaluation suggest that integrating AI into assessment can promote both technical fluency and ethical reflection when scaffolded appropriately…’
- Second, the manuscript lacks important contextual information about the participants. For a study focused on critical AI literacy and the integration of GenAI tools in assessment, it is essential to report details such as participants’ age, gender, academic background, and especially their prior experience and familiarity with GenAI tools for learning purposes. Such information is necessary to meaningfully interpret how students engaged with the task and reflected on the AI-generated content. Without this context, it is difficult to assess the validity and depth of their responses or to determine the broader applicability of the findings.
Due to the small academic cohort that participants were part of, we did not collect demographic information as reporting it would have a high risk of identifying individuals. We have however added some additional information about the cohort and restructured the participant section of methods to make it clearer how participants may have engaged with the task (lines 124-143)
‘The study involved twelve participants from the 2023–24 cohort of a postgraduate course at the Institute of Psychiatry, Psychology and Neuroscience, King’s College London, a Russell Group university in the United Kingdom. The student cohort comprised approximately 30 individuals. Most were in their early twenties and had entered the MSc programme directly after completing undergraduate studies, with around one in six being mature students returning to education after time in the workforce. A very small number were men. Approximately one-third were UK home students, while two-thirds were international, the majority of whom were from East Asia.
Eight students and four members of the teaching team took part in the study. The teaching staff included a Teaching Fellow, a Lecturer, and two Research Associates. All participants had recently completed or marked a summative assessment within the course. We considered the sample size adequate given the small cohort, the participatory nature of the research, and the principle of information power, which suggests that the more relevant information the sample holds, the lower the sample size needed (Malterud et al., 2016). In this study, participants were well positioned to inform the evaluation, having first-hand experience with the assessment and its development. They brought a range of expertise and experience with AI, from high digital literacy to limited prior use, as well as strengths in academic writing and assessment design. This ensured that the participatory methods supported shared ownership, practical relevance, and opportunities for innovation.’
- Third, the choice to use focus groups with students rather than individual interviews deserves an explanation. Since the AI-integrated task was completed individually, and only eight students were involved, it would have been feasible - and probably methodologically stronger, to conduct individual interviews. Focus groups can sometimes suppress divergent or critical views due to social dynamics. Individual interviews would have allowed for richer insights into each student’s unique experience with the task and the AI tools.
We have clarified this in the methods (lines 233-235):
‘This format was chosen to accommodate participant preferences and availability, while also helping to reduce power imbalances by providing students with a peer-supported setting in which to reflect on an assessment co-designed with researchers who were also their course instructors.’
- Lastly, although the authors includes a limitations paragraph that appropriately acknowledges the study’s confinement to a single institutional context, they do not address the significant limitation posed by the very small sample size. With only 8 student participants, the findings should be interpreted with considerable caution. I recommend that the authors revise their limitations section to explicitly acknowledge the small sample size and its implications for the validity and broader applicability of their findings.
In addition to the responses to #9, we have extended the limitations (lines 674-682) as follows:
‘Third, the study was conducted within a single institutional setting with a small cohort and an ensuing small sample size, which limits generalisability of the evaluation findings to other universities or international contexts with different AI access, policies, and pedagogical cultures. However, the study did not aim for statistical generalisability, but rather to explore the feasibility and acceptability in context, using participatory methods grounded in information power (Malterud et al., 2016). The broader goal was to model a co-design and evaluation approach that is transferable and could be adapted to different educational settings. The resulting assessment toolkit supports wider application, helping educators adapt AI-integrated assessments to their own institutional and disciplinary contexts.’
Results section
- In this section, participant quotes are presented – I recommend including basic identifiers alongside each quote. For example, participant codes such as (S1, Male) or (T3, Teaching Staff) to clarify who is speaking. This practice is common in qualitative research. Given the small number of participants, using consistent coding (e.g., S1–S8 for students and T1–T4 for staff) would be sufficient to maintain anonymity while allowing readers to better contextualize the quotes.
Thank you – we have updated Tables 5 and 6 and the results where relevant (lines 386-387) to include participant codes as suggested.
- Moreover, the paragraph in subsection 3.2 (rows 276-284) regarding marker accuracy and statistical testing requires clarification. It is currently unclear how the statistical comparison was justified and conducted given the extremely small number of cases. With such small cell sizes, the assumptions of the chi-square test are likely violated. Additionally, this statistical test is not mentioned in the Methods section.
This is a very good point, thank you for pointing out the oversight! We have added the below to the methods and results:
Methods (lines 263-267): ‘In addition to qualitative comparisons, we conducted a statistical analysis to compare how successful markers were at identifying assessments written by students in either the compliant or unrestricted conditions. Given the small sample size and expected cell counts of below five, we used the Fisher’s exact test rather than the chi‐square approximation (Howell, 2011), calculated using base R (R Core Team, 2025).’
Results (lines 435-439): ‘Markers correctly identified 6 out of 14 students in the compliant group (42.9%) compared to 3 out of 6 in the unrestricted group (50.0%). A Fisher’s exact test produced an odds ratio of 0.75, p = 1.00, indicating that marker accuracy did not differ meaningfully between the groups.’
Discussion section
- this section tends to reiterate results rather than critically interpret them in relation to the broader body of existing literature. There is limited engagement with recent empirical work in the field, particularly studies from 2024–2025 that have explored the integration of GenAI in higher education assessment. As I noted in my comments on the Introduction, this is a rapidly developing area, and the discussion would benefit from a stronger effort to position the study’s contributions within the evolving academic discourse. I encourage the authors to incorporate and reflect on recent and empirical studies that address similar pedagogical, ethical, and assessment-related challenges and innovations.
Thank you for this suggestion. We have now included more reflection in the Discussion that includes more recent and targeted references, including:
Pedagogical tool development: Gonsalves, C., 2024. Generative AI’s Impact on Critical Thinking: Revisiting Bloom’s Taxonomy. Journal of Marketing Education. 1-16.
Education transformation: Strielkowski, W., Grebennikova, V., Lisovskiy, A., Rakhimova, G., Vasileva, T., 2025. AI‐driven adaptive learning for sustainable educational transformation. Sustainable Development. 33, 1921-1947.
Academic authorship and scholarship: Kulkarni, M., Mantere, S., Vaara, E., van den Broek, E., Pachidi, S., Glaser, V.L., Gehman, J., Petriglieri, G., Lindebaum, D., Cameron, L.D., 2024. The future of research in an artificial intelligence-driven world. Journal of Management Inquiry. 33, 207-229.; Luo, J., 2024. A critical review of GenAI policies in higher education assessment: a call to reconsider the “originality” of students’ work. Assessment & Evaluation in Higher Education. 49, 651-664. https://doi.org/10.1080/02602938.2024.2309963.
Critical thinking: Larson, B.Z., Moser, C., Caza, A., Muehlfeld, K., Colombo, L.A., 2024. Critical thinking in the age of generative AI. Academy of Management Learning & Education. 23, 373-378; Zhai, C., Wibowo, S., Li, L.D., 2024. The effects of over-reliance on AI dialogue systems on students' cognitive abilities: a systematic review. Smart Learning Environments. 11.
- Additionally, a more critical reflection on the implications for practice and policy would strengthen the contribution of the manuscript.
Thank you for this interesting suggestion that we agree with. Across the reviewer feedback, it became clear that we had not fully articulated one of the central contributions of the manuscript, the participatory co-design and evaluation method itself. In response, we have revised the manuscript to more explicitly reflect the practical and policy implications of this approach. Throughout the revised text, we emphasise how participatory methods supported the development of an AI-integrated assessment that was feasible, acceptable, and pedagogically meaningful. We also clarify that the method is transferable to other educational settings and provide a practical toolkit to support educators in adapting and iteratively refining AI-integrated assessments in their own contexts. These culminate in the revised conclusion, where we include the method as a key outcome of the study and a contribution to current policy and practice in higher education assessment design.

Round 2
Reviewer 3 Report
Comments and Suggestions for Authors
Thank you for thoughtfully addressing my suggestions in the revised manuscript. The paper has improved significantly and reads much clearer now.